# Antimicrobial Peptide Arsenal Predicted from the Venom Gland Transcriptome of the Tropical Trap-Jaw Ant *Odontomachus chelifer*

**DOI:** 10.3390/toxins15050345

**Published:** 2023-05-18

**Authors:** Josilene J. Menk, Yan E. Matuhara, Henrique Sebestyen-França, Flávio Henrique-Silva, Milene Ferro, Renata S. Rodrigues, Célio D. Santos-Júnior

**Affiliations:** 1Laboratory of Molecular Biology, Department of Genetics and Evolution, Federal University of São Carlos (UFSCar), Rodovia Washington Luis, Km 235, São Carlos 13565-905, SP, Brazil; josilenemenk@gmail.com (J.J.M.); yematuhara@gmail.com (Y.E.M.); henrique.sebestyen@estudante.ufscar.br (H.S.-F.); dfhs@ufscar.br (F.H.-S.); 2Department of General and Applied Biology, Institute of Biosciences, São Paulo State University (UNESP), Rio Claro 01049-010, SP, Brazil; 3Laboratory of Biochemistry and Animal Toxins, Institute of Biotechnology, Federal University of Uber-lândia (UFU), Uberlândia 38400-902, MG, Brazil; 4Big Data Biology Laboratory, Institute of Science and Technology for Brain-Inspired Intelligence, Fudan University, Shanghai 200433, China

**Keywords:** antimicrobial peptides, prospection of peptides, venom peptides, venomous ants, stinging ants, neotropical ants

## Abstract

With about 13,000 known species, ants are the most abundant venomous insects. Their venom consists of polypeptides, enzymes, alkaloids, biogenic amines, formic acid, and hydrocarbons. In this study, we investigated, using in silico techniques, the peptides composing a putative antimicrobial arsenal from the venom gland of the neotropical trap-jaw ant *Odontomachus chelifer*. Focusing on transcripts from the body and venom gland of this insect, it was possible to determine the gland secretome, which contained about 1022 peptides with putative signal peptides. The majority of these peptides (75.5%) were unknown, not matching any reference database, motivating us to extract functional insights via machine learning-based techniques. With several complementary methodologies, we investigated the existence of antimicrobial peptides (AMPs) in the venom gland of *O. chelifer*, finding 112 non-redundant candidates. Candidate AMPs were predicted to be more globular and hemolytic than the remaining peptides in the secretome. There is evidence of transcription for 97% of AMP candidates across the same ant genus, with one of them also verified as translated, thus supporting our findings. Most of these potential antimicrobial sequences (94.8%) matched transcripts from the ant’s body, indicating their role not solely as venom toxins.

## 1. Introduction

The majority of insects from the order Hymenoptera (ants, bees, and wasps) have evolved stinging apparatuses which are used to deliver venom to their target [1]. Ants are arguably the most abundant venomous animals, presenting a ubiquitous distribution in terrestrial environments [2,3]. Their diversity is reflected in their venom, which is composed of a mixture of polypeptides, enzymes, alkaloids, biogenic amines, formic acid, and hydrocarbons, all of which are produced by the venom gland and stored in the animal’s abdomen under the venom reservoir [2]. The venoms of ant species in the Ponerinae, Myrmicinae, Pseudomyrmecinae, and Ecitoninae subfamilies are usually rich in proteins and peptides [1].

Ant venom peptides generally follow a structural pattern also found in the venom peptides of other hymenopteran insects, being short, linear, and polycationic [1]. These peptides have a high content of α-helices, which are responsible for cell lysis, hemolysis, releasing histamine, and neurotoxic and antimicrobial activity [1,3]. In addition to linear peptides, ant venoms may also contain structurally diverse peptides, such as dimeric forms and ICK-like inhibitor cystine knots [2].

There are many examples of peptide families found in ant venoms, such as poneratoxin, ponericin, ectatomin, and pilosulin, all with some degree of cytolytic or neurotoxic activity. Ant venom peptides with membrane-disturbing properties, e.g., ponericins, are usually associated with hemolytic, antimicrobial, and/or insecticidal properties. Their action as membrane-disrupting agents may permeabilize the cellular barriers, facilitating the neurotoxic peptides to achieve their targets. Poneratoxin and ectatomin peptides are toxic to vertebrate and invertebrate cells, mostly because of their neurotoxic abilities, which are related to the blocking of synaptic transmission by the modulation of voltage-gated sodium and calcium channels (for a review, see [2,3]).

The diversity of peptides found in ant venoms shows that they represent powerful weapons that, when injected into prey or predators, cause a wide variety of effects, ensuring the survival of ants in the majority of ecosystems in the world [1]. These peptides, when isolated, could be used in several biotechnological applications, such as the treatment of diseases (e.g., arthritis and infections), along with food conservation and pest control in agriculture [4].

Antimicrobial peptides (AMPs) derived from ant venoms can be used as an alternative antibiotic to treat infections caused by microorganisms [5]. This is mostly due to their broad-spectrum activity, targeting Gram-positive and Gram-negative bacteria, fungi, viruses, cancer cells, and parasites, allowing their use in a variety of disease treatments [6]. The major mode of action of AMPs is disrupting the target’s membrane shortly after binding to it. Moreover, in some cases, they are able to create membrane pores, ultimately causing cell lysis or leakage [7]. Alternatively, some AMPs target the synthesis of essential metabolites, such as DNA, RNA, and proteins, by inhibiting intracellular enzymes, which leads to the microorganism’s death [7].

One of these AMPs is bicarinalin, a peptide found in *Tetramorium bicarinatum* ants, which has shown activity similar to the already-used antibiotics for the treatment of Helicobacter pylori infections [8]. Another example of polyvalent AMPs is the pilosulins, marked as one of the major allergenic biologically active peptides in ants [9,10,11]. This class of peptides exhibits a wide range of activities, such as antimicrobial activity and hemolysis [12,13,14,15].

Given the possible applications of antimicrobial peptides and the fact that ants appear as promising sources of these molecules, we decided to investigate an unexplored neotropical trap-jaw ant. The ant genus *Odontomachus* [16] is in the subfamily Ponerinae and contains 77 species, 3 of which are fossils. These ants live in subtropical and tropical parts of the world and are simple to identify due to the design and kinetic action of their 180° jaw opening, which can produce clicks when closed [17]. *Odontomachus chelifer* [16] is a species that reproduces the individual foraging habits found in primitive ants of the Ponerinae subfamily and can be found in neotropical regions, such as the Brazilian Cerrado. Their nests are formed under layers of plant material or in abandoned galleries from other species of animals. *O. chelifer* feeding habits consist of an opportunist diet of dead arthropods found in the region they inhabit, and of termites killed with their powerful jaws and venom [17]. Using bioinformatics tools, we explored the peptide arsenal found in the transcriptome of the venom gland of the ant *O. chelifer*, also checking for evidence of translation and transcription in a sister species (*O. monticola*).

## 2. Results and Discussion

### 2.1. O. chelifer Body Preprocessing and Assembly

Illumina sequencing from the body of *O. chelifer* generated 63,012,700 reads. After preprocessing analysis, we discarded 15.7% of initial reads and obtained 52,469 contigs in a de novo assembly with Trinity. The assembly was 93.42% complete, according to our BUSCO analysis against Insecta_db10, while 4.02% of BUSCO groups were missing, and 2.56% were only partially recovered.

### 2.2. Potentially Secreted Peptides Found in O. chelifer

After the assembly of the venom gland transcriptome of *O. chelifer*, we obtained 125,123 proteins, of which we could predict 1,819 to have signal peptides. After filtering the mature sequences for those with lengths between 8 and 98 amino acids, the potential secretome harbored 1022 peptides (Figure 1, Appendix A). Only one peptide hit AntiFAM v.7 [18] with an incomplete domain and was discarded from downstream processes. The predicted secretome had an average disorder of 0.32 ± 0.27, with 366 out of the 1022 evaluated peptides being predicted to contain globular domains. Disorders above 0.45 indicated potential spurious protein predictions or disordered protein structures that mostly need help with folding [19,20]. Interestingly, a minority of the predicted secretome (282/1022 peptides) was assigned with this flag.

Only a small fraction (250/1022 peptides) of the mature sequences potentially secreted from the venom gland of *O. chelifer* was annotated with the Eggnog v.5 database [21], possibly due to the low homology between the ant peptides and the full-length proteins in the database. Despite the small representability of these annotated proteins, it was possible to observe the most represented category, that being the unknown metabolism class (58/250 peptides). In fact, the most abundant COG category with a defined function among the predicted secretome was the signal transduction mechanisms category (39/250 peptides), followed by the post-translational modification and chaperones category (28/250 peptides), the translation category (14/250 peptides), and the inorganic ion transport and metabolism category (11/250 peptides).

### 2.3. Predicted Antimicrobial Activity of Peptides That May Perform as Toxins in the Venom Gland from O. chelifer

Using a voting system with six different AMP classifiers, we predicted a total of 136 candidate AMPs (Figure 1, Appendix A) from the venom gland transcripts of *O. chelifer*, representing 112 non-redundant peptides. There was no significant difference (P_Mann-Whitney’s U_ = 0.92) between the disorder of the candidate AMPs and the residual secretome (the predicted secretome except for the predicted AMPs). However, the candidate AMPs tended to be more globular than the residual secretome peptides (odds ratio = 3.5, P_Fisher’s Exact_ = 4.6 × 10^−11^), which suggests that our candidate AMPs may be more soluble.

The predicted AMPs were almost twice as long as the average secretome, with medians of 60.5 and 28 (P_Mann-WhitneyU_ = 4 × 10^−19^, Figure 2A), respectively. We also found that the median length of AMPs (Q50 = 19) from public databases (e.g., DRAMP, APD3, dbAMP) is even shorter than the average secretome, indicating that candidate AMPs from *O. chelifer* may be naturally longer. Interestingly, longer AMPs usually cause the formation of pores in the bacterial membrane, while shorter ones promote the micellation of the membrane [22]. Candidate AMPs from *O. chelifer* were also more positively charged (pI_Q50_ = 10) than the total secretome (pI_Q50_ = 7.6, P_MannWhitneyU_ = 2.2 × 10^−10^), as seen in Figure 2B,C. Positive charges are essential to AMPs’ ability to bind the negatively charged membrane of bacteria [22], which implies a modulation of their antimicrobial activity based on the pH [22] and could indicate the best environment for their action, such as the venom itself or another body site. When analyzing the propensity of the predicted AMPs to bind other proteins via the Boman index [23], we observed no difference (P_MannWhitneyU_ = 0.39) between the predicted AMPs and the residual secretome (Figure 2D). This suggests that the majority of the peptides from the secretome, including the AMPs, perform protein–protein interactions, given the probability density having peaks at high values (>2.1).

Venoms are a complex cocktail of compounds and generally have functional diversity. This might be due to toxin multifunctionality, in which the same toxin achieves multiple functions, e.g., by target ubiquity. Cell membranes are probably the most ubiquitous potential targets for a toxin, often showing associated insecticidal, hemolytic, and/or antimicrobial properties [3,24]. In this context, we observed that candidate AMPs were more prone to cause hemolysis (odds ratio = 4.45, P_Fisher’s Exact_ = 2.3 × 10^−9^) than the residual secretome, which may be a plausible justification for their presence in the venom. One possible role of hemolysins in the venom of hymenopteran insects is pain induction in vertebrates as a defense against predators. These strategies are also described in bees and ants that use their venom only for defense [2].

We obtained weak hits (identity < 75%) with known sequences when annotating the candidate AMPs using several databases (e.g., UniProt, NCBI, CAMPR3). The only exception was the candidate TRINITY_DN3796_c0_g1_i1.p1, which was assigned to the family of defensins through Hidden Markov Models with the CAMPSign tool available at http://www.campsign.bicnirrh.res.in/ (accessed on 10 January 2023). When using other annotation tools, such as Blastp, we observed some strong hits (29/136) (identity > 90% and E-value < 1.0 × 10^−5^) against mostly full-length proteins from *O. brunneus* (22/29), indicating an incomplete assembly of those entities or a truncation in *O. chelifer*.

### 2.4. Phylogeny of AMP Candidates

We observed the phylogenetic relationship between the predicted AMPs and five reference antimicrobial peptides from other species (Figure 3). The reference AMPs were chosen based on the work of Zhang and Zhu [25], in which a comparative analysis of the genome of seven different ant species found 69 AMP-like genes that could fall into one of five AMP-like families. The predicted AMP sequences from *O. chelifer* were clustered based on the reference AMPs, revealing crustin-, hymenoptaecin-, defensin-, abaecin-, and ICK-like peptide clusters.

Crustins, first discovered in crustaceans, are AMPs targeting Gram-positive bacteria [25]. They are cationic and cysteine-rich, with a WAP (whey acidic protein) domain at the C-terminus. Two predicted AMP sequences from *O. chelifer* clustered with the Crustin4 from *Lasius niger*. The predicted AMP within the crustin-like peptide cluster that had the most (7) cysteine residues was chosen to have its tridimensional structure predicted (Figure 3A), sharing only 12.82% identity (Appendix A). Interestingly, its predicted structure is not similar to any other crustin-like peptide in UniProt, not even to crustin-like peptides with altered WAP domains (*Pacifastacus leniusculus*, PlCrustin2 A5A3L2), as previously described [26].

Invertebrate defensins have six conserved cysteine residues, which form three disulfide bonds that stabilize a complex arrangement of α-helices and β-sheets also known as a CSαβ motif. Although most of them show antimicrobial activity against Gram-positive bacteria, there are several defensins also targeting Gram-negative bacteria and/or fungi [27]. Twelve sequences of the predicted AMPs from *O. chelifer* clustered with the defensin-2 from *O. monticola*, including our only annotated AMP sequence that was modeled (TRINITY_DN3796_c0_g1_i1.p1) (Figure 3B), sharing 79.16% identity with the reference peptide (Appendix A). The CSαβ motif is clearly conserved in this AMP, similar to those of phormicin and heliomicin [27]. The N-terminal loop is fundamental for the activity of defensins, and the CSαβ motif is sensitive to its modification [28]. In ant defensins, this loop shows variable length, and this diversity may facilitate the evolution of novel antimicrobial effects in order to survive in different environments [25]. Interestingly, the N-terminal loop of the modeled defensin from *O. chelifer* appears to be longer than those previously described for other species.

Glycine-rich AMPs, such as hymenoptaecins, are found in a diverse number of hymenopterans [25] and usually target a broad spectrum of Gram-positive and Gram-negative bacteria [29]. Thirty-four predicted AMP sequences of *O. chelifer* clustered with the hymenoptaecin from *O. monticola*. Hymenoptaecins, similar to other glycine-rich peptides, do not adopt any particular tridimensional structure and are considered linear peptides [30]. Although the modeled hymenoptaecin from *O. chelifer* was mostly linear (Figure 3C), surprisingly, there were two visible α-helices in its structure, and it held 18.81% identity with the reference peptide (Appendix A). This might mean that this sequence is not a proper hymenoptaecin, but it could still be an AMP related to this group. 

ICK-type AMPs contain the ICK fold, which consists of an antiparallel triple-stranded β-sheet linked together by three disulfide bridges. This structural motif is found in venom toxins and in AMPs from many organisms, such as fungi, plants, mollusks, and arthropods [31]. Seven sequences of the predicted AMPs from *O. chelifer* clustered with the poneratoxin ICK-like AMP from *Dinoponera quadriceps*. Out of those, only two had at least six cysteine residues, and they had their secondary structure predicted using PSIPRED. Only one sequence showed at least three strands and was chosen to have its tridimensional structure modeled (Figure 3D), presenting 16% identity in relation to the reference AMP (Appendix A). This peptide supposedly has the necessary features for an ICK fold, being at least six cysteine residues and an anti-parallel triple-stranded β-sheet. However, the number of cysteine residues within the region of the triple-stranded β-sheet was inferior to that expected (Appendix A). This finding suggests that this peptide resembles ICK-type peptides, maybe keeping a functional closeness to this group. 

Proline-rich AMPs, such as abaecin, are linear cationic peptides that have a high percentage (>25%) of proline residues [32,33]. Their ability to translocate the bacterial membrane and kill the microorganism by targeting intracellular components is correlated with their amphipathic α-helical structures and the characteristic PXP/PXXP motif [34,35]. These non-lytic AMPs show a broad spectrum of bactericidal activity [34]. Fifteen sequences of the predicted AMPs from *O. chelifer* clustered with the abaecin from *O. monticola*. Out of those, only two peptides had >20% proline content, and their alignment with the abaecin from *O. monticola* revealed that they both possess the PXP/PXXP motifs (Appendix A), indicating a possible functional closeness. The sequence with the highest proline content (29.2%) shared an 18.75% identity with the reference AMP (Appendix A), and its tridimensional structure prediction showed a linear peptide (Figure 3E), just as expected for an abaecin-like peptide. Structured peptides are not always the more active ones, and the loops of a molecule are important for their activity and cytotoxicity [28].

### 2.5. AMPs May Also Compose the Immune System of O. chelifer

The majority of social ants, including *O. chelifer*, live in densely populated colonies, where food is accumulated in a warm and humid environment, facilitating the proliferation of pathogenic microorganisms [36]. The first line of host defense against these pathogens is the innate immune system, in which AMPs are essential elements to eliminate infections [25]. To further verify if candidate AMPs may also work in the innate immune system and not only as venom toxins, we mapped the transcripts from the venom gland of *O. chelifer* against the assembled transcripts of the ant body. We observed that 129 out of the 136 sequences of candidate AMPs hit transcripts from the body of the ant (Appendix A), with an average identity of 98.9 ± 3.5% and a maximum E-value of 1.7 × 10^−5^. It suggests that these candidate AMPs may also play a role in animal immunity, ultimately aiding in the maintenance of a healthy colony.

We also looked for evidence of the translation of predicted AMPs by searching the peptide sequences against a database of *O. monticola* peptides detected by LC-ESI-MS [37]. It was possible to retrieve one peptide matching the database (TRINITY_DN18722_c0_g1_i1.p2), suggesting that at least one of our candidate AMPs is actively translated. Moreover, that sequence was the only one in the entire predicted secretome to be detected. This is interesting due to the fact that most of the proteomes are not properly prepared to detect peptides. The resolution of the equipment mostly does not achieve the zone required to visualize such small proteins [38].

### 2.6. Conservation of Candidate AMP Transcription across the Genus Odontomachus

During the mapping of amino acid reads from *O. chelifer* with PALADIN [39], using only the mature sequences of the predicted secretome peptides as reference (Appendix A), we verified that 132 out of the 136 predicted AMPs (97%) were present in the original species. This suggests that 3% of the candidate AMPs are only generated during the assembly procedures or may represent potential artifacts. Furthermore, we could infer that the abundance of candidate AMPs is higher than the residual secretome (P_Mann-Whitney’s U_ = 3.4 × 10^−5^), indicating that the basal level of expression of these peptides may be higher than the background.

Lastly, the verification of these candidate AMPs across different species from the same genus could be an indication that these AMPs are conserved, and therefore should hold some function. We found a total of 85 out of 136 predicted AMPs in *O. monticola* by reverse-mapping the peptides against their amino acid reads. Only 23 out of the 85 candidate AMPs found across the two species presented a larger relative abundance in *O. monticola* in comparison to *O. chelifer*. These findings indicate the possibility that different peptides may perform tasks specifically related to the ant niche and habitat, suggesting a possible ecological role for these entities. 

## 3. Conclusions

We verified that the number of peptides predicted to be secreted in the venom gland is quite low (1022 out of 125,123). Additionally, we predicted that about 13.3% of these peptides may be antimicrobial under a rigorous quality-control setup and that this does not create a group of disordered proteins but globular proteins that tend to be more hemolytic than the average peptide in the secretome. This suggests that these peptides may be playing the role of toxins in the venom of *O. chelifer*. We also found evidence that the predicted AMPs are not only toxins, as they are present in the transcriptome from the *O. chelifer* body, which contains most of the candidate AMPs. Evidence that these peptides are also translated and transcribed in another species from the same genus strengthens our claims.

## 4. Materials and Methods

### 4.1. O. chelifer Venom Gland Transcriptome

We obtained the transcriptomic data for the *O. chelifer* venom gland that were previously generated by Guimarães and collaborators [40] and deposited in the GenBank SRA database. *O. chelifer* venom preparation and transcriptome analysis (RNA extraction, Illumina sequencing, quality control, preprocessing, and de novo assembly) are described in this article.

### 4.2. O. chelifer Body RNA Extraction and Sequencing

The total RNA from the body of *O. chelifer* was isolated using the TRIzol Reagent (Invitrogen, Waltham, MA, USA). The sample was macerated for approximately 2–3 min using a plastic pistil, and the extraction was performed according to the manufacturer’s instructions for the TRIzol reagent. The RNA was analyzed using a Nanodrop ND-1000 spectrophotometer (Thermo Scientific, Waltham, MA, USA), and its integrity was checked in a 1% agarose gel. The material was processed with RNAse-free DNAse I (Invitrogen) in a 20 μL final volume reaction containing 3 μg of RNA. The libraries were generated utilizing Illumina’s TruSeq small RNA library preparation kits. Poly (A) mRNA was isolated using oligo (dT) beads and through incubation with a fragmentation buffer 1x supplied with the Illumina small RNA sample preparation kit. The fragments were utilized as templates in reactions with SuperScript II reverse transcriptase (Invitrogen) and random hexamer primers to obtain the first-strand cDNA, followed by a reaction with DNA polymerase I and RNAse H (Illumina) to generate the double-strand cDNA. The fragments were end-repaired, the 3′ ends were adenylated, and adapters were ligated. Appropriate fragments (200 ± 30 bp) were isolated by agarose gel electrophoresis, purified, and enriched by polymerase chain reaction (PCR) using adapter-specific primers. The quality of the libraries was validated using an Agilent 2100 Bioanalyzer (Agilent Technologies, Santa Clara, CA, USA). Finally, the libraries were sequenced using HiSeq1000 (Illumina) in paired-end reads (2 × 100 bp), according to the manufacturer’s instructions.

### 4.3. O. chelifer Body Quality Control, Preprocessing, and De Novo Assembly

The FASTQC program in https://www.bioinformatics.babraham.ac.uk/projects/fastqc/ (accessed on 15 April 2022) was used to perform the quality control for the *O. chelifer* body transcriptome. A common phenomenon on the Illumina platform is known as a deviation in the GC content [41,42]. To solve this problem, the first 15 nucleotides of all reads were trimmed using the fastx_trimmer tool available in the FASTXToolkit at the webpage http://hannonlab.cshl.edu/fastx_toolkit/index.html (accessed on 15 April 2022). The last 4 bp were also removed using the same program. SeqyClean [43] was used to filter low-quality bases (Phred Q score < 30). De novo assembly was performed with Trinity v2.14.0 [44] using default parameters. Aiming to provide a quantitative and comprehensive overview of the level of completeness for our assembly, we used the Benchmarking Universal Single-Copy Orthologs (BUSCO) v5 program [45]. The assembly was compared with a predefined set of Insecta databases and sequences were categorized as “complete, single copy”, “complete, duplicated copy”, “fragmented”, or “missing”, according to the aligned sequence.

### 4.4. Obtaining Coding Protein Regions from O. chelifer Transcriptome

A total of 50,220 transcripts from the *O. chelifer* venom gland were used to obtain the coding protein regions using TransDecoder v5.5.0 [46] in the OmicsBox v3.0.25 (BioBam, Valencia, Spain) [47] platform. Proteins were filtered by size to a maximum of 200 amino acids.

### 4.5. Prediction and Annotation of the Peptides Composing the O. chelifer Venom Gland Secretome

We worked with 125,123 predicted proteins from the transcripts assembled from the venom gland of *O. chelifer*. Their signal peptides were predicted using Razor v.1 [48], leaving only peptides with predicted cleavage sites and falling in the range of 8 to 98 amino acids. This precise length threshold was defined to allow the machine learning downstream operations that sometimes require peptides in these ranges. The remaining 1022 peptides, edited to their mature sequences, were searched against the AntiFAM v.7 [18] database with HMMER v.3.3.2 [49] to eliminate spurious sequences. The hemolytic activity of the secreted peptides was assessed using Macrel v.1.2.0 [50], and the disorder along with globular domains was computed using IUpred3.0 [20], the latter not using smoothing procedures. The predicted secretome was annotated using eggNOG-mapper v.2.1.10 [51] to assess COG classes and trace a metabolism profile.

### 4.6. Prediction of Potentially Secreted Antimicrobial Peptides

The predicted peptides from the secretome of the venom gland of *O. chelifer* were submitted to different systems to predict antimicrobial peptides (AMPs): (i) those based in neural networks: AI4AMP [52], AMPlify [53], and AMP Scanner v2 [54]; and (ii) those based in random forests: amPEPpy [55], ampir using the model for mature sequences [56], and Macrel v.1.2.0 [50]. The final result for each peptide was obtained through the voting system, where peptides assigned as candidate AMPs by three or more systems were kept and the others discarded. In order to verify if candidate AMPs would be more likely to cause hemolysis—a plausible justification for their presence in the venom—a Fisher’s Exact test was applied in the counts of candidate AMPs, and the other peptides from the secretome were organized according to their hemolytic activity. The odds ratio was then computed.

Candidate AMPs were annotated using online services from UniProt release 2022_05 [57] and CAMPR3 [58], namely the CAMPSign web server available at http://www.campsign.bicnirrh.res.in/ (accessed on 10 January 2023) with its Hidden Markov Models (HMM) database of AMP families, and the full Blast mode against several different AMP databases using a maximum E-value of 1 × 10^−5^. The annotation also proceeded locally using Blastp [59] against the NCBI database release 254 [60] and parsing results to those with an identity ≥ 90%, no gap opens, alignments from the start of the peptide, and, again, a maximum E-value of 1 × 10^−5^, keeping only the best hit for each sequence.

### 4.7. Cross-Mapping Experiments

We inferred the presence of these predicted AMPs in the body of *O. chelifer* through the mapping of transcripts from the *O. chelifer* venom gland against the assembled transcripts from the body using BlastN [61] in the OmicsBox platform. Evidence of the translation of predicted AMPs was gathered by searching the peptide sequences against a database of *O. monticola* peptides detected by LC-ESI-MS [61]. Using PALADIN [39], we mapped the potential peptides secreted against the venom gland transcriptome of *O. chelifer* in amino acid reads terms to verify the abundances of potentially antimicrobial peptides versus the residual secretome from the venom gland of *O. chelifer*. Using PALADIN again [39], we mapped the amino acid reads from the gland transcriptome of *O. monticola* against the predicted antimicrobial peptides of *O. chelifer*, inferring not only their presence across different species but quantifying their cross-species abundance.

### 4.8. Statistical Testing

Differences between proportions were verified using Fisher’s Exact test and the differences between medians were assessed using the Mann–Whitney U test, both implemented in the Scipy 1.9.1 package [62] from Python 3.9. The significance threshold was established as *p* < 0.05. Graphical views of distributions were rendered using the Seaborn [63] and Matplotlib [64] packages.

### 4.9. Predicted AMP Clustering and Phylogeny

Based on the work of Zhang and Zhu [25], we included different AMP sequences from five families largely present in ant taxa in the predicted AMPs to compare their features and possibly infer their activities/mechanisms. These included a defensin-2 (A0A348G6A9), hymenoptaecin (A0A348G6C2), and abaecin (A0A348G6C5) from *Odontomachus monticola*; a poneritoxin ICK-like (P0DSL6) from *Dinoponera quadriceps;* and a crustin 4 (A0A0J7L9H5) from *Lasius niger*. The additional sequences had their signal peptides removed prior to downstream operations. All the peptides were then clustered using CDHIT [65] at 100% identity and 90% overlap of the shorter sequence (parameters: “-c 1 -aS.9 -G 0 -l 5”). Then, the sequences were aligned using MAFFT [66] in the “auto” mode with the posterior alignment trimmed using ClipKIT [67]. The phylogenetic tree was produced with FastTree2 [68], using pseudocounts to correct for the sparsity, estimating the transitional rates using gamma distribution, and estimating the maximum likelihood from the Le Gascuel (LG) [69] model implemented. To estimate the feature distribution and closeness among the potentially secreted and antimicrobial sequences, the same 22 biochemical features (e.g., pI, charge, and Boman index) used by Macrel [50] for the AMP prediction were calculated and compared as density functions. After alignments, and keeping most of the reference’s domains, the sequences clustering to references in the phylogenetic tree followed downstream procedures. ESPript [70] from https://espript.ibcp.fr (Accessed on 15 April 2023) was used to edit and visualize some alignments. Secondary structure predictions were generated using PSIPRED 4.0 [71]. Peptide tridimensional structure predictions were made with ColabFold v1.5.2: AlphaFold2 using MMseqs2 [72]. 

## Figures and Tables

**Figure 1 toxins-15-00345-f001:**
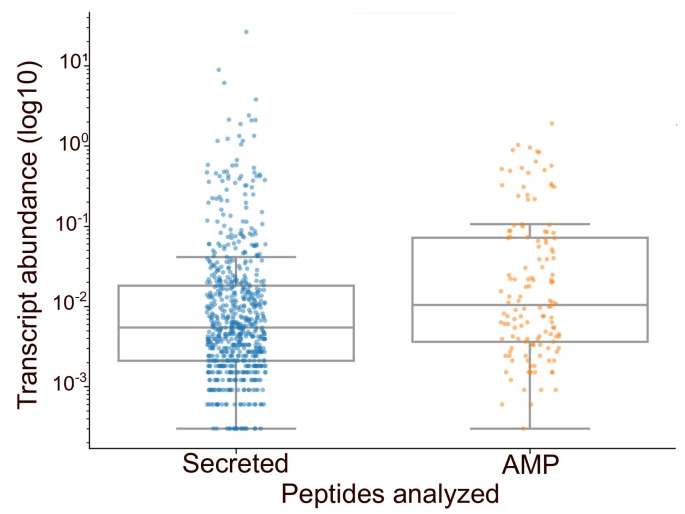
Graphical analysis of the number of peptides found in the total secretome of *O. chelifer*. The graphics compare the abundance of peptides predicted in the total secretome with the number of predicted AMPs in the same secretome.

**Figure 2 toxins-15-00345-f002:**
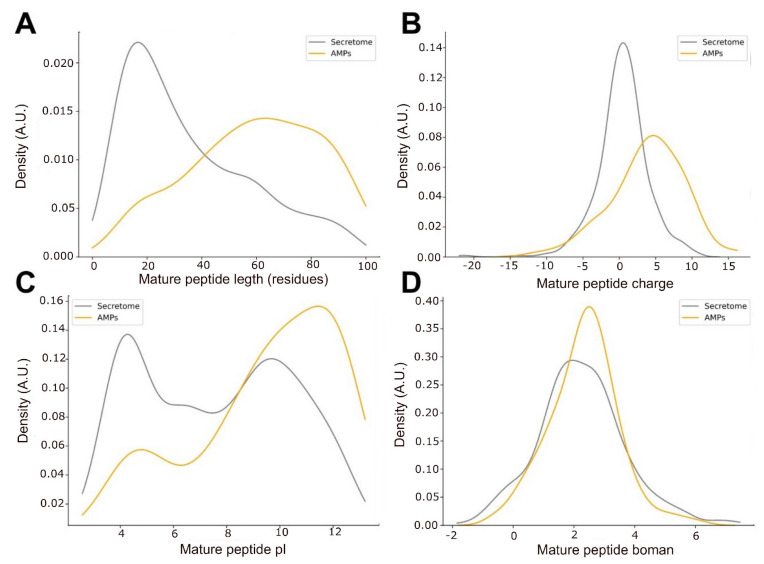
Comparison of the biochemical characteristics of the predicted AMPs and the total secretome from the *O. chelifer* ant venom. The four graphics contain the comparison between the (**A**) length, (**B**) charge, (**C**) pI, and (**D**) Boman index of the predicted AMPs and the total secretome.

**Figure 3 toxins-15-00345-f003:**
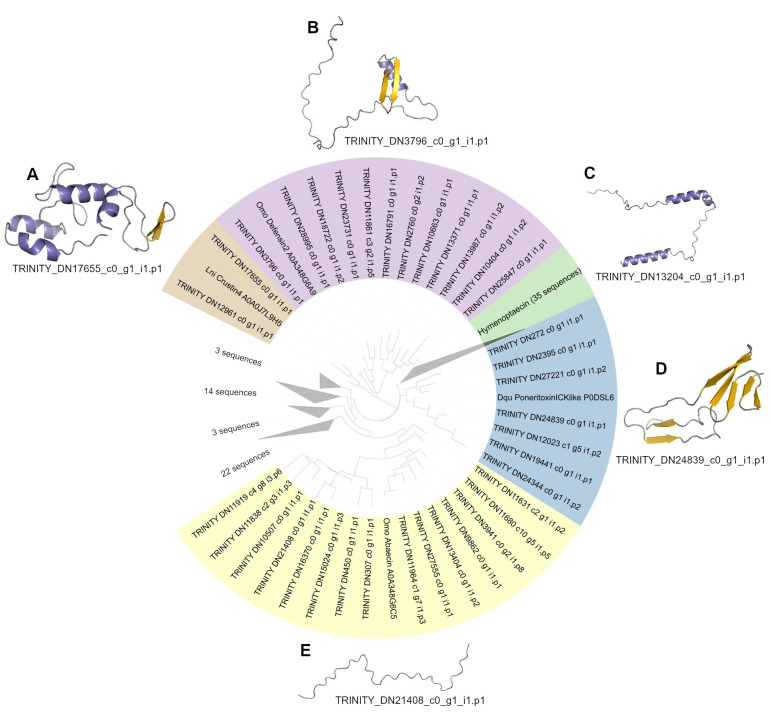
A phylogenetic tree of the predicted antimicrobial peptides (AMPs) from *O. chelifer* and five reference AMPs from other ant species. The groups are clustered based on the five reference AMP sequences. Sequences from outgroups are left uncolored. Tridimensional structure models for sequences belonging to different clusters of (**A**) crustin-like, (**B**) defensin-like, (**C**) hymenoptaecin-like, (**D**) ICK-type, and (**E**) abaecin-like peptides are also available, following the same color pattern: purple for helix, yellow for strand, and gray for turns. The sequences used for structure prediction are listed below each model. Detailed alignments of all sequences from each group of the tree, including source tissue and their signal peptides, are presented in Appendix A.

## Data Availability

Peptides from the secretome and potentially antimicrobial molecules predicted in this study are available as Appendix A in this paper. Reads from the *O. monticola* venom gland are available in the SRA database (BioProject—PRJDB5661, BioSample—SAMD00078338, SRA—DRP004401, and run—DRR093871). The reads from *O. chelifer* are also available in the SRA databases for the body (BioProject—PRJNA895366, BioSample—SAMN33553712, SRA—SRR23674720) and the venom gland (BioProject—PRJNA895366, BioSample—SAMN31503287, and SRA—SRR22084210).

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
