# Peer review of "Antimicrobial Peptide Arsenal Predicted from the Venom Gland Transcriptome of the Tropical Trap-Jaw Ant Odontomachus chelifer"

_toxins, 2023, doi:10.3390/toxins15050345_

Round 1

Reviewer 1 Report

The manuscript “Antimicrobial peptides arsenal predicted from the venom gland transcriptome of the tropical trap-jaw ant Odontomachus chelifer” submitted by unknown authors to the Toxins presents in-silico analysis of antimicrobial peptides-related transcripts identified from venom gland of the neotropical trap-jaw ant Odontomachus chelifer, using next-generation sequencing. On my opinion, this study would be of interest to specialists dealing with proteinaceous toxins and antimicrobial drug development.

I really like research design, analysis, and presentation in Section 2.3. However, there are a few major comments.

1. The authors wrote that they had taken the raw sequence data from the O. chelifer transcriptome (SRA: SRR22084210). Following Reference 40, I found that “SRA Experiment SRX18064352 is not public Data”. The BioSample SAMN31503287 is missing, as is BioProject PRJNA895366. Therefore, it is difficult to evaluate the data if they are absent. There are no links to O. chelifer body transcriptome numbers (SRA, BioProject, BioSample).

2. I have a few questions related to the choice of library preparing kits and the experiment itself. First, the authors used “Poly (A) mRNA isolated by oligo (dT) beads” and generated the transcriptome “utilizing Illumina's TruSeq Small RNA Library Preparation Kits”. This kit uses total RNA or purified small RNA as input. It does not make fragmentation and adenylation. Also, about “following pair-ending (2 X 100 bp), adenylation and ligation…” (Line 316), it means that the authors made mate pair sequencing with short- or long-insert pair-end DNA library. If it is so, then it is required detailed description.

3. It is highly recommended to show peptide alignments (with signal peptide sequences, not just mature peptides) of all five AMP-like families, add information on the degree of similarity between these peptides. It is extremely inconvenient to use a supplementary table to understand the structural diversity of the peptides. I also recommend specifying the source (tissue) of sequences in the alignments.

4. I would like to recommend the authors to present a phylogenetic tree (Figure 3) as a graphic abstract, because the construction of a phylogenetic tree is based on the cysteine pattern (or what?) does not make much sense; their homology is very low, branching is mostly unreliable. The authors should pay more attention to the establishment of phylogenetic relationships within peptide families, to identify orthologs, paralogs and multigene family members. The authors should discuss peptide gain and lost comparing with data from other ant transcriptomes.

Reviewer 2 Report

This manuscript describes the set of potential antimicrobial peptides found in the venom gland transcriptome of the ant Odontomachus chelifer by the use of machine learning-based methods. I’m unfamiliar with these methods, so was unable to judge this aspect of data generation. However, I found no obvious flaws in the paper. I do recommend that the authors carefully check the language of the manuscript to correct errors (e.g. line 36: ‘in the venom reservoir’; lines 69-71: species names need to be italicized).

Round 2

Reviewer 1 Report

I am satisfied with the revised manuscript in its current form. The authors have fine-tuned their manuscript. Further corrections are not required, the manuscript may be accepted for publication.